**Subject Area:**
cellular biology/cognition/neuroscience/biotechnology/developmental biology

neurodegenerative disease, Alzheimer's disease, brain organoid, Huntington's disease, induced pluripotent stem cells, Parkinson's disease

**Author for correspondence:**
Hung-Chih Kuo
e-mail: kuohuch@gate.sinica.edu.tw

†These authors contributed equally to this work.

A contribution to the special collection commemorating the 90th anniversary of Academia Sinica.

# Opportunities and challenges for the use of induced pluripotent stem cells in modelling neurodegenerative disease

Yi-Ying Wu[1,†], Feng-Lan Chiu[1,†], Chan-Shien Yeh[1,†] and Hung-Chih Kuo[1,2,3]

[1]Institute of Cellular and Organismic Biology, and [2]Genomics Research Center, Academia Sinica, Taipei 11529, Taiwan, Republic of China
[3]Graduate Institute of Medical Genomics and Proteomics, College of Medicine, National Taiwan University, Taipei, Taiwan, Republic of China

Y-YW, 0000-0002-1512-3308

Adult-onset neurodegenerative diseases are among the most difficult human health conditions to model for drug development. Most genetic or toxin-induced cell and animal models cannot faithfully recapitulate pathology in disease-relevant cells, making it excessively challenging to explore the potential mechanisms underlying sporadic disease. Patient-derived induced pluripotent stem cells (iPSCs) can be differentiated into disease-relevant neurons, providing an unparalleled platform for *in vitro* modelling and development of therapeutic strategies. Here, we review recent progress in generating Alzheimer's, Parkinson's and Huntington's disease models from patient-derived iPSCs. We also describe novel discoveries of pathological mechanisms and drug evaluations that have used these patient iPSC-derived neuronal models. Additionally, current human iPSC technology allows researchers to model diseases with 3D brain organoids, which are more representative of tissue architecture than traditional neuronal cultures. We discuss remaining challenges and emerging opportunities for the use of three-dimensional brain organoids in modelling brain development and neurodegeneration.

## 1. Introduction

Neurodegenerative diseases are often characterized by progressive atrophy of neurons and tissue, which corresponds to a loss of neuronal function and results in impaired cognition and/or movement. Each specific neurodegenerative disease preferentially affects a defined population of neurons, leading to distinctive age-related clinical profiles [1]. However, many neurodegenerative diseases share common molecular features that precede neuronal death and dysfunction, including mitochondrial dysfunction, axonal damage and abnormal protein aggregation [2,3]. Aberrant processing and aggregation of misfolded proteins cause complex and distinctive pathophysiological profiles in several neurodegenerative proteinopathies. Some well-known hallmarks of these diseases include Amyloid $\beta$ (A$\beta$) plaques and phosphorylated Tau (pTau)-containing tangles in Alzheimer's disease (AD) [4], $\alpha$-synuclein-associated Lewy bodies in Parkinson's disease (PD) [5] and mutant huntingtin (Htt)-containing inclusion bodies in Huntington's disease (HD) [6]. These aggregated proteins may act via loss-of-function or gain-of-toxicity mechanisms to cause neuronal axon damage and cellular vulnerability. Thus, clearance of neurotoxic aggregation is a major focus of phenotypic assays for drug development, which includes methods to monitor autophagy-lysosomal network function, chaperone-mediated folding and clearance, and ubiquitin-proteasome protein degradation [3]. Many pathogenic mutations are associated with protein processing and/or aggregation. For example, HD is a monogenic disease

caused by a polyglutamine expansion in mutant Htt, which causes the protein to aggregate. Likewise, several known pathogenic mutations in AD are associated with A$\beta$ production. However, the majority of AD and PD cases are idiopathic, which makes exploring disease mechanisms very difficult without access to damaged tissue in the patient's nervous system. Post-mortem brain tissues have provided essential pathological information for each disease, but it is not suitable for identifying the biological changes during initial stages of disease. Furthermore, transgenic animals are valuable models for phenotypic and preclinical testing during drug development, but microenvironment and species differences may be major reasons that transgenic animals have been largely unable to sufficiently recapitulate disease phenotypes. Current approaches to drug discovery have not delivered effective therapeutics to reduce neurodegeneration in AD [7], and other neurodegenerative suffer from a lack of therapeutic options. Thus, the current models may be complemented by access to patient-derived disease-relevant neural cell types, greatly aiding preclinical drug evaluation for neurodegenerative disease.

Recent advances in the ability to reprogram patient somatic cells into inducible pluripotent stem cells (iPSCs) have provided a novel means to generate disease-relevant cells for *in vitro* disease modelling [8,9]. Human iPSC technology was launched by Yamanaka and colleagues when they first introduced the transcription factors, OCT4, SOX2, KLF4 and c-MYC, to somatic cells, generating a novel method for producing stem cells [10]. In principle, human iPSCs can differentiate into any cell type of human body; thus, patient iPSCs can provide a source of cells that harbour a precise constellation of genetic variants, which is associated with pathogenesis in the appropriate microenvironment. As such, iPSCs are often used in well-established models of human disease, including both developmental and adult-onset diseases, in the form of either two-dimensional (2D) cell cultures or three-dimensional (3D) organoids [9,11–16]. Importantly, cells derived from patient iPSCs have been shown to recapitulate phenotypes of various human neurodegenerative diseases, including AD [17,18], amyotrophic lateral sclerosis [19,20], HD [21] and fragile X syndrome [22]. Also, improvements in iPSC culture and the development of robust differentiation protocols have made it possible to carry out phenotype-based drug screening in iPSC-derived disease-target cells [11,18,20,23]. Expandable iPSCs can give rise to a large number of disease-related cells, providing an excellent opportunity for large-scale drug testing [9]. However, several technical considerations should be taken into account when applying this approach. For example, one key issue is that variability in the phenotypes of iPSC lines from individual patients necessitates a large cohort of lines to eliminate misleading pathological mechanisms or drug effects. In order to address this issue, the use of current gene-editing technology has allowed researchers to standardize genetic background by using isogenic control lines [24,25]. Thus, coupling of gene editing technologies with patient-derived iPSCs has enabled the generation of a set of genetically defined human iPSC lines for disease modelling [24]. Another hurdle for modelling disease with iPSC-derived cells is that the maturity of derived neurons and differentiation time required for phenotypes to emerge may be variable across iPSC lines [26]. This variability issue can be addressed by the use of multiple well-characterized iPSC lines and isogenic controls. Moreover, for most diseases of ageing, multiple or chronic treatments are required to promote the expression of disease-associated phenotypes in cellular models [27–33]. This challenge is significant, but may be addressed in many cases by the use of long-term 3D organoid cultures. These complex structures provide unique human organ-like tissue that is amenable to long-term culturing for disease modelling. The self-organizing capability of iPSCs can recapitulate several key features of human cortical development, including progenitor zone organization, neurogenesis, gene expression and distinct human-specific outer radial glia cell layers [34]. Furthermore, the complex structures promote disease pathogenesis by accelerating neuronal differentiation and maturation, providing excellent laboratory models for human neurodegenerative disease.

The great potential for the use of iPSC technology in developing treatments for human disease is evident [25]. In this review, we provide an overview of iPSC technology in modelling neurodegenerative diseases of the central nervous system (especially AD, PD and HD), including methods for differentiating disease-relevant neurons, important findings in drug development, and current trends for improving treatment of neurodegenerative disease. We also discuss the use of iPSC-derived 3D brain organoids to study the central nervous system and current findings from this technology with regard to neurological diseases. The advantages and disadvantages of iPSC 3D organoid modelling and potential new treatments for neurodegenerative diseases are highlighted.

## 2. iPSC-based disease modelling and drug discovery in Alzheimer's disease

AD is the most common form of dementia in the elderly, affecting more than 40 million people worldwide [35]. The primary gross pathology of the disease is brain volume reduction and hippocampal degeneration, while the pathological hallmarks are extracellular A$\beta$ plaques and aggregation of hyperphosphorylated tau in neurofibrillary tangles. The result of these pathological processes is that AD patients suffer progressive memory impairment and acute cognitive dysfunction during late-stage disease [4]. Experimental therapies that target A$\beta$ deposition have been thus far unsuccessful in clinical trials. Although current medications, which include cholinesterase and N-methyl-D-aspartate (NMDA) inhibitors, cannot stop neuronal loss, the drugs may lessen and stabilize cognitive defects [4].

Both genetic and environmental factors are likely to be involved in AD pathogenesis. Most genetic forms of AD result in disease onset before age 60 and are termed early-onset or familial AD. By contrast, the most pervasive form of AD is idiopathic with increasing incidence after 65 years of age; this category is denoted late-onset AD or sporadic AD [36]. Importantly, familial AD-associated gene mutations are all involved in A$\beta$ production and include amyloid-$\beta$ precursor protein (*APP*) [37], presenilin1 (*PSEN1*) [38] and presenilin 2 (*PSEN2*) [39]. Theoretically, these genetic mutations cause Amyloid $\beta$ 1–42 (A$\beta$42) production and result in extracellular A$\beta$ aggregation. However, the causes of sporadic AD are poorly understood. A few genetic factors, including apolipoprotein E (*APOE*) [40], sortilin-related receptor (*SORL1*) [41] and SMI-1 (*SMI1*) [42], were reported to be associated with late-onset AD. It has also been

royalsocietypublishing.org/journal/rsob    Open Biol. 9: 180177

royalsocietypublishing.org/journal/rsob    Open Biol. 9: 180177

suggested that complex interactions between genetic factors and other host factors, such as diabetes mellitus, hypertension and obesity, play important roles in the aetiology of late-onset AD [43]. Nevertheless, the specific causes and mechanisms underlying the occurrence and progression of sporadic AD are still elusive.

iPSCs have been widely used to explore disease pathogenesis associated with both inherited monogenetic mutations and sporadic AD (table 1). Since multiple neuronal types are susceptible in AD, protocols have been developed to generate different subtypes of forebrain neurons from AD-iPSCs for use as *in vitro* models of disease (table 2). The accumulation of A$\beta$ peptide, which is produced from APP via sequential cleavage by $\beta$-secretase and $\gamma$-secretase, is thought to be a causative factor of AD pathology [36,54], and the elevation of A$\beta$ was found to result from pathogenic mutations in *APP*, *PSEN1* and *PSEN2*. *PSEN1* encodes a subunit of $\gamma$-secretase that releases soluble APP from the cellular membrane. Several mutations in presenilin1 (*PSEN1*) have been identified, including A79 V [45], Y115C, M146I [38], A264E [39,48], G265C [47], G384A [49], T449C [47], S169del [48] and a single nucleotide deletion in intron 4 [38]. Additionally, all of these mutations have been shown to elevate A$\beta$42 secretion in AD-iPSC derived cortical neurons. *PSEN1*(P117R) was associated with reduced neurite length and susceptibility to inflammatory response [46]. It was also shown that hyper-phosphorylation of microtubule-associated protein Tau (pTau) occurs in AD-iPSC-derived neurons carrying *PSEN1* mutations [47,48]. Moreover, AD-iPSC-derived neural progenitors with S169del or A264E mutations were shown to exhibit low rates of proliferation and high rates of apoptosis [48]. Similarly, *PSEN2* encodes the catalytic subunit of $\gamma$-secretase and is involved in amyloid-$\beta$ precursor protein (APP) cleavage. *PSEN2*(N141I) was associated with increased A$\beta$42 secretion and decreased action potential in neurons derived from patent iPSCs [39,50,51]. Mutations in *APP* involving V717 (either V717I or V717 L) and *APP* duplication have been shown to cause the elevation of A$\beta$, Tau and pTau in AD-iPSC-derived forebrain neurons and astrocytes [37,38,52]. The elevated Tau expression was reduced by A$\beta$ antibody treatment [38]. Furthermore, *APP*(E693del) was associated with accumulation of intracellular A$\beta$ oligomers and susceptibility to stress response in forebrain neurons [27]. In general, A$\beta$42 secretion was reduced in familial AD iPSC-derived neurons after treatment with $\beta$- or $\gamma$-secretase inhibitors, including BSI-IV, compound E, compound W or DAPT. NSAID or imidazole-based modulators of $\gamma$-secretase activity were also shown to be effective at reducing A$\beta$42 secretion [45].

The majority of AD cases are diagnosed as sporadic or polygenic, suggesting that AD is most often a multifactorial disease that arises from genetic variants and environmental factors [24,43]. Genome-wide association studies (GWASs) have identified numerous genetic variants associated with sporadic AD, such as GRB2-associated binding protein (*GAB2*), galanin-like peptide (*GALP*), piggyBac transposable element derived 1 (*PGBD1*), tyrosine kinase, non-receptor 1 (*TNK1*) and clusterin (*CLU*, also known as apolipoprotein J) [55]. *APOE* encodes Apolipoprotein E, a cholesterol carrier lipoprotein in the brain, which has several isoforms or alleles. The *APOE4* allele was the first gene risk factor to be identified for sporadic AD and is still the most significant. Patients with *APOE4* have elevated risk compared with those carrying *APOE3*, while *APOE2* is considered to be a protective allele [56]. *APOE4* codes for APOE(C112R), which has altered binding affinity towards lipoproteins and A$\beta$ [57]. In human iPSC-derived forebrain neurons, the *APOE4* allele was associated with high levels of pTau, A$\beta$ secretion and GABAergic neuron degeneration [40,44,53]. These AD-related phenotypic events were reduced by treating cells with the *APOE4* structure corrector, PH002 [40]. In *APOE3/E4* neurons, apigenin, an anti-inflammatory drug, showed neuroprotective effects by reducing Ca$^{2+}$ signalling frequency and caspase 3/7-mediated apoptosis [46]. In *APOE3/E4* forebrain cholinergic neurons, neurotoxicity was increased when cells were treated with ionomycin, and cell viability was reduced while calcium was elevated upon glutamate treatment [44]. SORL1 is functionally associated with directing APP to endocytic pathways. A certain genetic variation in the *SORL1* 5′ region is known as a gene-risk factor for sporadic AD, and it has been shown that this variant reduces expression of *SORL1* in AD-iPSC-derived neurons. Additionally, BDNF treatment cannot induce *SORL1* expression to reduce A$\beta$ secretion by contrast with wild-type *SORL1* carriers [41]. *BMI1* encodes polycomb complex protein BMI1, and is associated with transcriptional repression of several genes through Ring1 mediated E3-mono-ubiquitin ligase activity. In cortical neurons generated from sporadic AD patient-derived iPSCs, BMI1 was downregulated and associated with AD phenotypes including A$\beta$ secretion/extracellular deposition, Tau phosphorylation and neuronal degeneration. Mechanistically, BMI1 was associated with transcript repression of microtubule-associated protein tau (*MAPT*) and destabilization of glycan synthase kinase-3$\beta$ (GSK-3$\beta$) and p53. Thus, several drugs that target these mechanisms were effective for reducing phenotypes [42]. Also, the AD-associated phenotypes, such as elevations in A$\beta$ secretion and Tau phosphorylation, and activation of GSK-3$\beta$ were identified in neurons generated from sporadic AD patient-derived iPSCs [17,27,58]. These phenotypes could be reduced by treating AD-iPSC-derived neurons with the $\beta$-secretase inhibitors $\beta$Si-II and OM99-2; however, $\gamma$-secretase inhibitors (Compound E and DAPT) had no rescue effects on the sporadic AD-iPSC-derived neurons [17].

As AD-iPSC-derived neurons can recapitulate pathological features of disease, these cells provide a promising platform for the identification of potential drug targets and further drug development. Drug screening has been carried out in AD-iPSC-derived neurons to search for compounds that can reduce A$\beta$ toxicity-mediated cell death. Accordingly, cyclin-dependent kinase 2 inhibitors were identified as agents that can robustly reduce A$\beta$ neurotoxicity [18]. Furthermore, six pharmaceutical compounds were also identified for the ability to lower A$\beta$ production. Among these compounds, a combination of bromocriptine, cromolyn and topiramate showed potent anti-A$\beta$ effects in AD-iPSC-derived neurons [49].

# 3. iPSC-based disease modelling and drug discovery in Parkinson's disease

PD is one of the most common adult-onset neurodegenerative diseases, affecting 1% of people over the age of 60 worldwide [59]. The core pathology of PD involves selective loss of A9-type dopaminergic neurons that project from the

Table 1. Alzheimer's disease modelling based on patient iPSCs.

| iPSC genotypes | main finding | β or γ-secretase inhibitors | other treatments | differentiated cell type | cell markers | refs |
|---|---|---|---|---|---|---|
| **familial Alzheimer's disease** | | | | | | |
| PSEN1 (A246E, M146J, Y115C, intronic single nucleotide deletion) | Aβ42↑ Aβ42/40↑, carboxypeptidase activity↓ | compound E/DAPT/ E2012: Aβ↓; compound W: Aβ42↓ Aβ42/40↓ | none | cortical neurons, basal forebrain cholinergic neurons | MAP2, TUJl, CTIP/TBR1, CHAT/ VACHT/ Nkx2.1 | Yagi et al. [39]; Duan et al. [44]; Moore et al. [38] |
| PSEN1(P79V) | Aβ40↓ Aβ42/40↑ | high-level NSAID-based γ-secretase modulators: Aβ42/40↓ | none | neurons | MAP2, TUJ1 | Mertens et al. [45] |
| PSEN1(P117R) | neurite length↓; susceptibility for inflammatory stress↑ | none | apigenin: neurite length↑; stress susceptibility↓ | neurons | MAP2 | Balez et al. [46] |
| PSEN1 (G265C, T449C) | Aβ40↑ Aβ42↑, APP↑ APP-CTF↑, pTau↑, activated GSK-3β↑ stress susceptibility↑; PSEN1 mutations: Aβ42/40↑ | none | none | cholinergic neurons, dopaminergic neurons, glutamatergic neurons, GABAergic neurons | MAP2, TUJ1, VACHT, TH, VGLUT1/2, GAD2/1 | Ochalek et al. [47] |
| PSEN1 (A246E, S169del) | premature NPCs: proliferation↓ apoptosis↑; Neurons: Aβ42↑ Aβ42/40↑ pTau↑ | none | PSEN1 knockdown: rescues NPC prematuration | neurons | MAP2, TUJ1 | Yang et al. [48] |
| PSEN1(G384A) | Aβ40↓ Aβ42↑ Aβ42/40↑ | BSI-IV or GSM: Aβ↓ Aβ42/40↓ | 6 pharmaceutical compounds: Aβ↓, Aβ42/40↓ | cortical neurons | MAP2, TBR2, SATB2, VGLUT1 | Kondo et al. [49] |
| PSEN2(N141I) | Aβ42↑, Aβ42/40↑, AP number↓ First AP height↓ | compound E: Aβ↓; compound W: Aβ42↓ Aβ42/40↓ | insulin: Aβ42↓, calcium flux↑; correction of PSEN2-N141I: restores all deficits | basal forebrain cholinergic neurons | MAP2, TUJ1, FOXG1, p75/CHAT/ VACHT/ Nkx2.0 | Yagi et al. [39]; Ortiz-Virumbrales et al. [50]; Moreno et al. [51] |

**Table 1.** (*Continued.*)

| iPSC genotypes | main finding | β or γ-secretase inhibitors | other treatments | differentiated cell type | cell markers | refs |
|---|---|---|---|---|---|---|
| *APP* [E693Δ, APP-V717 L] | *APP-E693Δ*: Aβ↓; *APP-V717 L*: Aβ42↑ Aβ42/40↑; *APP-E693Δ* | BSI-IV: Aβ↓ sAPPβ↓ Aβ oligomer↓, stress responses↓ | DHA: stress responses↓ | cortical neurons | MAP2, TUJ1, TBR1/SATB2 | Kondo et al. [49] |
| *APP(K724N)* | Aβ40↓ Aβ42↑ Aβ42/40↑ | high-level NSAID-based GSM: Aβ42/40↓ | none | neurons | MAP2, TUJ1 | Mertens et al. [45] |
| *APP (V171I)* | Aβ↑, Aβ42/40↑, sAPPβ↑, tTau↑pTau↑; GABAergic neurons: the highest Aβ level; astrocytes: secreting high level of Aβ | DAPT: Aβ↓ sAPPβ↓ | Aβ specific antibody: Tau↓ | forebrain neurons | MAP2, TUJ1, TBR1/CUX1, VGLUT1 | Muratore et al. [37] |
| *APP (V717I, APP^DP)* | *APP-V717I*: Aβ40↓ Aβ42↑ Aβ42/40↑; *APP^DP*: Aβ↑; Both: tTau↑ pTau↑ | DAPT/E2012: Aβ↓, altered pTau and Tau level; DAPT: APP-C83/99↑ | none | cerebral cortex neurons | TUJ1, CTIP/ TBR1 | Moore et al. [38]; Liao et al. [52] |
| *APP^Dp* | Aβ40↑, pTau(S231)↑, activated GSK-3β↑, early endosome accumulation | compound E/DAPT: Aβ40↓; BSI-II/ OM99-2: Aβ40↓ pTau↓ active GSK-3β↓ | none | glutamatergic neurons, GABAergic neurons, cholinergic neurons | MAP2, TUJ1, VGLUT1, GAD67/ GABA | Israel et al. [17] |
| **sporadic Alzheimer's disease** | | | | | | |
| *APOE3/E4* | Aβ42↑ Aβ42/40↑; neurotoxic susceptibility↑ calcium transient↑ | compound E/DAPT: no effects | none | basal forebrain cholinergic neurons | MAP2, CHAT/ VACHT/ Nkx2.1 | Duan et al. [44] |
| *APOE3/3* or *E4/E4* | Aβ↑ pTau↑ | none | ApoE4 structure corrector: Aβ↓ pTau↓ | GABAergic neurons | MAP2, TUJ1, GABA | Wang et al. [40] |
| *APOE4* | Aβ42↑ pTau↑; synapse number↑; Aβ uptake ↓; microglia-like cells: morphology changes; astrocytes: cholesteral accumulation | none | none | astrocyte, microglia-like cells, neurons | MAP2, S100β, Iba1 | Lin et al. [53] |

(*Continued.*)

**Table 1.** (Continued.)

| iPSC genotypes | main finding | β or γ-secretase inhibitors | other treatments | differentiated cell type | cell markers | refs |
|---|---|---|---|---|---|---|
| SORL1-5′-end SNPs | BDNF in normal neurons: SORL1↑ Aβ↓; BDNF in sAD: no effects | none | none | glutamatergic neurons, GABAergic neurons, cholinergic neurons | MAP2, TUJ1, VGLUT1, GAD67/GABA | Young et al. [41] |
| BMI1 | BMI1↓, Aβ42↑ Aβ oligomer↑ Tau↑ p-Tau↑, apoptosis in GABAergic neurons↑, amyloid plaques and p-Tau tangles formation, PSD-95 and synaptophysin↓, GSK-3β↑ p-p53↑, MAPT expression↑ | BSI-IV, DAPT, GSM-XXII: Aβ↓ p-Tau↓ apoptosis↓ | caffeine, DNA-PK inhibitor or p53 inhibitor: amyloid↓ p-Tau↓; ATM/ATR inhibitor or ATM inhibitor: apoptosis↓; GSK-3β inhibitor: amyloid↓ p-Tau↓ apoptosis↓ | GABAergic neurons, glutamatergic neurons, cholinergic neurons | MAP2, TUJ1, GABA, VGLUT1, ChAT | Flamier et al. [42] |
| unkown | neurite length↓; Aβ42↑, apoptosis or cytotoxicity from oxidative/nitrosative stress↑, hyper-excitability↑ | none | apigenin: neurite length↑; apoptosis and hyper-excitability↓ | neurons | MAP2 | Balez et al. [46] |
|  | Aβ40↑ Aβ42↑, APP↑ APP-CTF↑, pTau↑, activated GSK-3β↑ stress susceptibility↑ early endosome accumulation | compound E/DAPT: Aβ40↓; BSI-II/OM99-2: Aβ40↓ pTau↓ active GSK-3β↓ | none | cholinergic neurons, dopaminergic neurons, glutamatergic neurons, GABAergic neurons | MAP2, TUJ1, VACHT, TH, VGLUT1/2, GAD2/1 | Ochalek et al. [47]; Israel et al. [17] |
|  | intracellular Aβ oligomer↑, stress responses↑ | BSI-IV: Aβ↓ sAPPβ↓ Aβ oligomer↓, stress responses↓ | DHA: stress responses↓ | cortical neurons | MAP2, TUJ1, TBR1/SATB2 | Kondo et al. [49] |

BSI: β-secretase inhibitor; GSM: γ-secretase modulator.

**Table 2.** Differentiation protocols for AD relevant neurons.

| methods for neuronal induction | supplement for NPCs generation | supplement for differentiation | neuronal markers | differentiated cell type | refs |
|---|---|---|---|---|---|
| **EB formation** | | | | | |
| no bFGF, with or without dual SMAD inhibitors/dorsomorphin | neurosphere formation: B27, insulin, progesterone, FGF2 | B27, insulin, progesterone | MAP2, TUJ1 | none | Yagi et al. [39] |
| | neurosphere formation: N2, SB431542, dorsomorphin | B27, BDNF/GDNF/NT-3 | MAP2, TUJ1, TBR1/SATB2 | cortical neurons | Kondo et al. [27] |
| | EB attachment and rosette formation: N2 → NPC expansion: N2, B27, FGF2/EGF2 | N2, B27, cAMP | MAP2, TUJ1, CTIP/TBR1 | cortical neurons | Mertens et al. [45]; Moore et al. [38] |
| | EB attachment and rosette formation: N2, heparin, SHH → neurosphere formation: N2, B27, heparin, cAMP, IGF1 | N2, B27, BDNF/GDNF/IGF1, cAMP | MAP2, TUJ1, TBR1, CUX1, VGLUT1 | forebrain neurons | Muratore et al. [37]; Liao et al. [52] |
| | EB attachment and rosette formation: N2, heparin → neurosphere formation: N2, B27, heparin, bFGF | N2, B27, SHH/FGF-8 → N2, B27, SHH/FGF-8, AA, cAMP, BDNF/GDNF/IGF-1 | MAP2 | none | Balez et al. [46] |
| | EB attachment and rosette formation: N2, B27, heparin, bFGF/EGF → neurosphere formation: N2, B27, heparin, bFGF/EGF | N2, B27, BDNF/GDNF | MAP2, TUJ1, GABA | GABAergic neurons | Wang et al. [40] |
| | neurosphere formation: N2, heparin, bFGF/EGF → neuceofection of Lhx8/Gbx1-IRES-EGFP: SHH/FGF8 | sorting GFP+ cells: B27, bFGF, NGF, cytosine arabinoside | MAP2, CHAT/ VACHT/ Nkx2.1 | basal forebrain cholinergic neurons | Duan et al. [44] |
| **NPC induction** | | | | | |
| RA, sodium bicarbonate. | neurosphere formation: N2, heparin, bFGF/EGF | N2, B27, BDNF/GDNF/CNTF | MAP2, TUJ1 | none | Yang et al. [48] |
| N2, Noggin, SB431542, sodium bicarbonate | EB attachment and rosette formation: N2, B27, bFGF | | | | |
| N2, B27, LDN, SB431542, bFGF | N2, B27, EGF/bFGF | N2, B27, AA | MAP2, TUJ1, VACHT, TH, VGLUT1/2, GAD2/1 | cholinergic neurons, dopaminergic neurons, glutamatergic neurons, GABAergic neurons | Ochalek et al. [47] |

(Continued.)

**Table 2.** (Continued.)

| methods for neuronal induction | supplement for NPCs generation | supplement for differentiation | neuronal markers | differentiated cell type | refs |
|---|---|---|---|---|---|
| N2, LDN, Noggin | N2, B27 | B27 | MAP2, TUJ1, GABA, VGLUT1, ChAT | GABAergic neurons, glutamatergic | Flamier et al. [42] |
| PA6 co-cultured, Noggin, SB431542 for NPCs isolation (CD184$^+$, CD15$^+$, CD44$^-$, CD271$^-$) | N2, B27, bFGF | N2, B27, BDNF/GDNF, cAMP for isolating neurons (CD24$^+$, CD184$^-$ CD44$^-$) | MAP2, TUJ1, VGLUT1, GAD67/GABA | neurons, cholinergic neurons | Israel et al. [17]; Young et al. [41] |
| B27, doxycycline hydrochloride | none | B27 | MAP2, TBR2, SATB2, VGLUT1 | cortical neurons | Kondo et al. [27] |
| LDN, SB431542, SAG, purmorphamine | p75$^+$ NPCs isolation : Brainphys medium, B27 neurosphere formation: Brainphys medium, B27, NGF/BDNF | Brainphys medium, B27, NGF/ BDNF | MAP2, TUJ1, FOXG1, p75/CHAT/VACHT/ Nkx2.1 | basal forebrain cholinergic neurons | Ortiz-Virumbrales et al. [50]; Moreno et al. [51] |

AA, ascorbic acid; RA, retinoic acid; NPC, neural progenitor cell.

substantia nigra (SN) in the midbrain to the dorsal striatum [60]. The pathological hallmark of PD is Lewy bodies, which consist of intra-neuronal aggregates of the synaptic protein α-synuclein [61]. PD clinical manifestations include motor deficits, such as tremor, rigidity, akinesia and postural instability. At present, there is no cure for PD, but dopamine (DA) replacement [62–64] or deep brain stimulation may be prescribed for relief of motor symptoms [65].

Both genetic and environmental factors are likely to be involved in PD pathogenesis. About 10% of PD cases are caused by inherited genetic mutations; most of the responsible genes are involved in regulation of mitochondrial function and oxidative stress, including SNCA (α-synuclein) [5,29], PARK2 (Parkin) [66], PINK1 (PTEN-induced kinase 1) [67], PARK7 (protein deglycase DJ-1), LRRK2 (leucine-rich repeat kinase) [67–69] and ATP13A2 (ATPase type 13A). GWASs have also identified SNPs and triplication of SNCA [70], LRRK2, GBA1 (β-Glucocerebrosidase), MAPT (microtubule-associated protein tau) [59] and GAK (cyclin G-associated kinase) [71–73] as being highly associated with sporadic PD [74,75]. Certain environmental factors have also been shown to be associated with PD pathogenesis, including exposures to certain pesticides, herbicides, heavy metals and bacteria. For modelling PD in animals, parkinsonism may be experimentally induced by disrupting mitochondrial function with MPTP (1-methyl-4-phenyl-1,2,3,6-tetrahydropyridine) [76] or 6-OHDA (6-hydroxydopamine) [69,77].

Modelling PD with disease iPSCs has been widely used to explore pathogenesis associated with inherited monogenetic mutations as well as sporadic PD (summary in table 3). SNCA encodes α-synuclein and is the first gene that was linked to PD. Although the function of α-synuclein is not well understood, α-synuclein aggregation in Lewy bodies is a major pathological phenotype of PD. SNCA(A53T) was shown to be associated with α-synuclein aggregation and Lewy body-like deposition in dopaminergic neurons derived from PD-iPSCs. Moreover, neuronal death was caused by mitochondrial dysfunction due to nitrosative and oxidative stress [32]. Triplication of SNCA leads to doubling of α-synuclein expression in dopaminergic neurons derived from PD-iPSCs [78], and this genetic aberration is linked to cell death with mitochondrial swelling in cortical neurons [79]. Furthermore, SNCA (A53T and triplication) was associated with ER and nitrosative stress in PD-iPSC-derived cortical neurons [29]. Oligomeric α-synuclein was identified in cortical neurons harbouring SNCA triplication and was associated with neuronal death. Mechanistically, oligomeric α-synuclein selectively induced oxidation of the ATP synthase β-subunit, leading to permeability transition pore-associated cell death [79]. Although the pathological role of α-synuclein structure is contentious, Lewy bodies are known to consist mainly of fibrils and are composed of insoluble α-synuclein β-sheet structures [82]. Autosomal dominant LRRK2 (leucine-rich repeat kinase 2) encodes a protein containing multiple-functional domains including a protein kinase, GTPase and protein-interacting regions. Mutations in LRRK2 have been correlated with both familial and sporadic PD. LRRK2(G2091S) was associated with upregulation of α-synuclein protein, elevated expression of key oxidative stress-response genes and mitochondrial dysfunction in dopaminergic neurons that were derived from PD-iPSCs. GW50764 (an LRRK2 kinase inhibitor) prevented

**Table 3.** Parkinson's disease modelling based on patient iPSCs.

| gene mutation | main findings | differentiated cell type | marker | main components | days | refs |
|---|---|---|---|---|---|---|
| **familial Parkinson's disease** | | | | | | |
| SCNA (A53T) | decreased α-synuclein ratio of tetramer: monomer | neurons | MAP2 | AA, Dorsomorphin, FGF2, Noggin | 55 | Dettmer et al. [5] |
| | monomer initiate pathology and further neurotoxity and inclusion | | | | | |
| | nitrosative /oxidative stress caused S-nitrosylation of MEF2C | DA neuron | OTX2, FOXA2, LMX1A, TH, NURR1, GIRK2 | AA, cAMP, BDNF, CHIR99021, DAPT, FGF8, GDNF, LDN193189, Noggin, PMA, SB431542, SHH, TGF-β | 25 | Ryan et al. [32] |
| | inhibition of MEF2C-PGC1α contribute to mitochondria dysfunction and cell death | | | | | |
| | gene-environment interaction(GxE) involved in PD pathogenesis | | | | | |
| | small-molecule high-throughput screening identify MEF2C-PGC1α as a therapeutic target | | | | | |
| SCNA (A53T or triplication) | early phenotype: nitrosative and ER stress | cortical neurons | TUJ1 | BDNF, cAMP, GDNF | 31 | Chung et al. [29] |
| | accumulation of ER degradation and the substrate | | | | | |
| SCNA (triplication) | upregulated α-synuclein expression | DA neurons | TUJ1, TH, LMX1, NURR1, DAT | AA, BDNF, Dorsomorphin, Dkk1 blocking ab, Noggin, FGF2, GDNF, SB43152, SHH, TGF-β, Wnt1 | 20 | Devine et al. [78] |
| | α-synuclein monomer improves ATP synthase efficiency | cortical neurons | TUJ1 | BDNF, GDNF, LDN, SB431542 | 70–90 | Ludtmann et al. [79] |
| | oligomers selectively increase oxidation of ATP synthase and mitochondrial lipid peroxidation | | | | | |
| | opening permeability transition pore, mitochondrial swelling and cell death | | | | | |

(Continued.)

**Table 3.** (*Continued.*)

| gene mutation | main findings | differentiated cell type | marker | main components | days | refs |
|---|---|---|---|---|---|---|
| *LRRK2* (G2019S) | upregulation of α-synudein; elevated stress response induced by hydrogen peroxide; sensitive to 6-OHDA, MG132 and hydrogen peroxide | DA neurons | TH, PITX, FOXA2, NURR1 | AA, BDNF, cAMP, FGF8, GDNF, Noggin, SB431542, TGF-β | 55 | Nguyen *et al.* [69] |
| *LRRK2* (G2019S) | impairment of nuclear envelop organization; defective self-renewal and neuronal differentiation; sensitive to MG132 induced cell death | NSC | SOX2, PAX6, Nestin | AA, BDNF, CHIR99021, Comp. E, cAMP, Dorsomorphin, FGF8, GDNF, LIF, PMA, SB431542, TGF-β | 7 | Liu *et al.* [68] |
| *LRRK2* (G2019S), *PINK1* (Q456X) | mitochondrial dysfunction; cell vulnerabilities; reduce cell death by treating antioxidants: coenzyme Q10 and rapamycin, reduce cell death by treating LRRK2 kinase inhibitor GW5074 | DA neurons | TH, TUJ1, FOXA2 | AA, BDNF, cAMP, FGF8, GDNF, Enzo, Noggin, Retinoic acid, TGF-β, WNT1 | 53 | Cooper *et al.* [67] |
| *PINK1*(Q456X), *PARK2* (V324A) | progerin induces DA neurons ageing; dendrite degeneration, loss of TH cells; Lewy-body precursor | DA neurons | TH, TUJ1, mAP2 | BDNF, CHIR99021, cAMP, DAPT, FGF8, GDNF, LDN193189, Noggin, PMA, SB431542, SHH, TGF-β | 20–32 | Miller *et al.* [33] |
| *PARK2* (R42P, exon 3 deletion, exon 3,4 deletion, R275 W) | reduced capability of differentiation into DA neurons; altered mitochondrial: cell volume fraction | DA neurons | TH, TUJ1, FOXA2 | BDNF, cAMP, FGF2, FGF8, GDNF, LIF, SHH, TGF-β | 28 | Shaltouki *et al.* [66] |
| *PARK2* (exon3,5 deletion or exon3 deletion) | enhanced oxidative stress; increased dopamine release; decreased dopamine uptake | DA neurons | TH, EN-1, FOXA2, DAT, VMAT2, NR1, MAP2 | AA, BDNF, cAMP, FGF2, FGF8, GDNF, Noggin, SB431542, TGF-β, SHH | 70 | Jiang *et al.* [80] |
| **sporadic Parkinson's disease** | | | | | | |
| *SCNA* (SNPs) | brain specific transcription factors, EMX2 and NKX6-1, regulated expression of *SCNA* | neurons | TUJ1, MAP2, TH, NeuN, GFAP, vGlut1 | AA, Dorsomorphin, FGF2, Noggin | 25–31 | Soldner *et al.* [70] |

(*Continued.*)

**Table 3.** (*Continued.*)

| gene mutation | main findings | differentiated cell type | marker | main components | days | refs |
|---|---|---|---|---|---|---|
| *GBA* (N370S and L444P) | elevated α-synuclein expression; decreased glucocerebrosidase activity; lower dopamine level, increased monoamine oxidase B (MAO-B) expression; overexpression of wild-type GBA and MAO-B inhibitors are potential treatment | DA neurons | TH, MAP2, TUJ1, | BDNF, CHIR99021, cAMP, DAPT, FGF8, GDNF, LDN193189, Noggin, PMA, SB431542, SHH, TGF-β | 34 | Woodard *et al.* [72] |
| *GBA1* (N370S) | decreased glucocerebrosidase activity and protein associated with autophagic and lysosomal defects; cell death associated with increased neuronal $Ca^{2+}$-binding protein 2 | DA neurons | TH, TUJ1, FOXA2, NURR1, GIRK2, VMAT2 | AA, BDNF, cAMP, CHIR99021, DAPT, FGF8, GDNF, LDN193189, PMA, SB431542, SHH, TGF-β | 20–34 | Schondorf *et al.* [71] |
| unknown | T lymphocytes induce cell death through IL-17R and NF-κB activation; IL-17 antibody, secukinumab, rescued neuron death | DA neuron | TH, TUJ1 | AA, CHIR99021, cAMP, FGF8, GDNF, LDN193189, PMA, SB431542, TGF-β | 56 | Sommer *et al.* [81] |

**cell replacement therapy in Parkinson's disease based on human iPSC**

| models | main findings | cell types | marker | main components | days | refs |
|---|---|---|---|---|---|---|
| rat | transplantation method for DA neurons in striatum of PD rat without damaged host striatum; transplanted DA neurons containing long term survival; DA neurons projected into host striatum | DA neuron | transplantation: NCAM, TH, TUJ1; graft validation: TH, NCAM, GIRK2, L1 | AA, BDNF, cAMP, FGF8, GDNF, SHH, TGF-β | 42 | Hargus *et al.* [62] |

royalsocietypublishing.org/journal/rsob   Open Biol. **9**: 180177

(*Continued.*)

**Table 3.** (*Continued.*)

| cell replacement therapy in Parkinson's disease based on human iPSC | | | | | |
|---|---|---|---|---|---|
| models | main findings | cell types | marker | main components | days | refs |
| primate (*Macaca fascicularis*) | improve movement after transplanted CORIN$^+$ cells | DA progenitor/DA neurons | transplantation: CORIN | AA, A83-01, BDNF, cAMP, CHIR99021, LDN193189, GDNF, PMA | 12 | Kikuchi et al. [63] |
| | extended DA neuron neurites in host striatum | | graft validation: FOXA2, NURR1 | | | |
| | various neuron grafted and survival rates in host | | | | | |
| | no tumour formation from CORIN sorted cells for 2 years' study | | | | | |
| monkey | improved motor behaviour after transplantation with LRTM1$^+$ cells | DA progenitor/DA neurons | transplantation: LRTM1, FOXA2, LMX1A | A83-01, CHIR99021, FGF8, LDN193189, PMA | 10 | Samata et al. [64] |
| | DA neuron survival; no tumour formation | | Graft validation: GIRK2, NURR1, TH | | | |

NSC, neuron stem cell; DA neurons, dopaminergic neurons; TH, tyrosine hydroxylase; A83-01, TGF$\beta$ kinase/activing receptor-like kinase (ALK 5) inhibitor; Enzo: Smoothened agonist; SHH, Sonic Hedgehog; 6-OHDA, 6-hydroxydopamine; AA, ascorbic acid; PMA, purmorphamine.

neuronal cell death, implying that blocking LRRK2 kinase activity may be a valuable drug mechanism [67]. Moreover, PD-iPSC-derived dopaminergic neurons were sensitive to apoptosis after exposure to stressors including hydrogen peroxidase, MG132 and 6-OHDA. This enhanced sensitivity is consistent with our current understanding of early PD phenotypes [69]. In neural stem cells, LRRK2(G2091S) was associated with defective self-renewal and neuronal differentiation [68]. LRRK2(G2091S) was also surprisingly correlated with the disintegration of nuclear envelope, which was associated with ageing in other human diseases [30,68,83]. This iPSC-based research suggested that the nuclear pore structure may be useful for early diagnosis of PD and could become a therapeutic target. Mitochondrial abnormalities are a commonly studied mechanism of dysfunction in PD research. As such, a recessively inherited early-onset form of PD is caused by a mutation in PINK1 (PTEN-induced putative kinase 1) encoding a mitochondria-localized kinase, which accumulates on the outer membrane of damaged mitochondria [84]. Similar to mutations in LRRK2, mitochondrial dysfunction was associated with cell death in PINK1(Q456X)-carrying dopaminergic neurons. Cell viability could be rescued by antioxidant reagents, coenzyme Q10 and rapamycin [67]. In dopaminergic neurons with induced ageing by overexpression of progerin (a truncated nuclear envelope protein laminin A), the PINK1(Q456X) and PARK2(V324A) mutants exhibited neuron-specific neuromelanin accumulation. In the same neurons, other PD-associated phenotypes were recapitulated, such as loss of tyrosine hydroxylase (converts L-tyrosine to L-DOPA), Lewy-body inclusions and enlarged mitochondria [33]. This report was consistent with the previous findings that impairment of nuclear pore structure is associated with PD, and the overexpression of progerin provides a tool to accelerate the ageing process and study late-onset age-related PD for drug development. Another PD-related gene, PARK2 (Parkin), is an E3 ubiquitin ligase that targets mitochondria with PINK1 accumulation for degradation [84]. Mutations in PARK2 are associated with an autosomal recessive early-onset familial PD and are correlated to loss tyrosine hydroxylase-positive dopaminergic neurons [66]. An exon deletion that results in the loss of Parkin expression leads to increased oxidative stress, reduced DA uptake and increased spontaneous DA release in dopaminergic neurons derived from PD-iPSCs. These observations suggest that Parkin is involved in controlling DA neurotransmission and suppressing DA oxidation in human midbrain dopaminergic neurons [80].

Apart from monogenic inherited PD, several genes have been identified as risk factors for sporadic PD. Mutated GBA1 is a well-validated risk factor for PD [85]. GBA1 encodes the lysosomal enzyme β-glucocerebrosidase, which is involved in glycolipid metabolism. Mutations in GBA1 (N370S and L444P) were correlated to lowered β-glucocerebrosidase activity and α-synuclein accumulation in dopaminergic neurons derived from PD-iPSCs [72]. The substrate of β-glucocerebrosidase, glucosylceramide, was accumulated in cells with either GBA1 mutations or defects in autophagic and lysosomal machinery. In GBA1(N370S)-carrying dopaminergic neurons, DA synthesis and release were reduced. Monoamine oxidase B (MAO-B) expression was upregulated, and the inhibitor, rasagiline, rescued DA regulation [72]. Moreover, NECAB2 (neuronal calcium-binding protein 2) was increased, causing the dysregulation of neuronal calcium

homeostasis and increasing the vulnerability of cells to stress from cytosolic calcium elevation [71,72]. In an iPSC-based platform, MAO-B inhibitors or overexpression of wild-type GBA1 were shown to be potential therapeutic strategies for PD treatment [72]. Recently, the adaptive immune system was suggested to be associated with PD after researchers detected higher Th17 frequency in blood and upregulated T lymphocytes in post-mortem tissues. In co-culture with activated T lymphocytes, cell death was induced via upregulation of IL-17 receptor and NF-κB activation in dopaminergic neurons derived from sporadic PD-iPSCs. Blockade of IL-17 by the IL-17 antibody, secukinumab, provided a potential method for rescue of neuronal cell death [81].

Disease modelling with 2D iPSC-derived cultures may not be ideal due to a lack of complexity and neuronal immaturity, which are especially disadvantageous for modelling adult-onset sporadic diseases. Recently developed methods to generate brain organoids may help to create complex 3D models of midbrain tissue from iPSCs. These midbrain organoids are attractive models for mechanistic studies and drug discovery for PD due to the inclusion of well-characterized neurons, astroglia and oligodendrocytes [86]. Indeed, the genetic signatures of the brain and intestinal organoids derived from PD-iPSCs carrying the LRRK2(G2019S) mutation were altered compared to controls. Although further work is needed to elucidate the molecular pathology caused by the PD-associated mutations, these studies demonstrate the utility of the 3D organoid platform for PD research [87]. 3D brain or midbrain organoids will allow the exploration of multiple factors that contribute to PD and broaden the potential targets for drug development to neuron-adjacent cells.

# 4. iPSC-based disease modelling and drug discovery in Huntington's disease

HD is an autosomal dominant, fatal, progressive neurodegenerative disorder which is monogenic with exonic CAG repeat in the huntingtin (HTT) gene. The expanded CAG repeat encodes a polyglutamine tract that causes a toxic gain of function and leads to preferential death of GABAergic projection neurons in the striatum. Typically, HD symptoms typically manifest in midlife with motor deficits. Healthy individuals have an average number of CAG repeats ranging from 10 to 35, and HD patients have 36 or more expanded CAGs. Notably, CAG repeat length is highly correlated to disease severity and the onset age. Most importantly, there is no cure for HD, and the only treatments available are for the management of symptoms [88].

Since striatal medium spiny neurons (MSNs) are the major susceptible cell type in HD, many differentiation protocols have been developed to generate MSNs from hESCs and human iPSCs (table 4). Although it is now possible to derive highly enriched MSNs from HD patient-derived iPSCs for disease modelling, the recapitulation of HD-relevant phenotypes, including neuronal degeneration and aggregation of mutant huntingtin (mHtt) protein in HD-iPSC-derived neurons often requires the addition of other cellular stressors. For example, it has been shown that HD-iPSC-derived MSNs cultured in vitro have elevated levels of caspase activity upon growth factor withdrawal [91,95,96], hydrogen peroxide treatment [89,90,95] or glutamate stimulation [95]. Moreover, the formation of mHtt aggregates in the HD-iPSC-derived

**Table 4.** Differentiation protocols for HD relevant neurons.

### HD iPSC-derived striatal neurons

| CAG repeat length | differentiated cell type | markers | differentiation duration | patterning factor | stressor | main finding | compound testing | refs |
|---|---|---|---|---|---|---|---|---|
| 43 | MSNs | TUJ1/MAP2, GABA, GAD65, Calbindin, or DARPP32 | 16 weeks | N2B27 media with bFGF→B27 media | hydrogen peroxide | $H_2O_2$-induced more DNA damage and cell death in HD-MSNs; protective effect via $A_{2A}R$ antagonist | $A_{2A}R$ antagonist: SCH5826; PKA inhibitor: H-89 | Chiu et al. [89] |
| 40, 42, 47 | GABA MSN-like neurons | TUJ1, GAT1, or DARPP32 | 40–80 days | B27 media with BDNF, Forskolin | proteasome inhibitor (MG132) | MG132 induced more EM48$^+$ cells; enhanced SOC activity in HD-MSNs and reduced by EVP4593 | Quinazoline derivative: EVP4593 | Nekrasov et al. [90] |
| 28, 33, 60, 180 | striatal neurons | TUJ1, MAP2, GABA, DARPP32, Bcl11B | 72 days | NIM with SHH, DKK1, BDNF→NIM with cAMP, Valproic acid, BDNF | hydrogen peroxide; autophagy inhibitor (3-MA); glutamate; BDNF withdrawal | increased striatal neurons death | none | HD-iPSC Consortium [28] |
| 72 | HD-NSCs and striatal neurons | TUJ1, GABA, Calbindin, DARPP32 | none | NDM with SHH, DKK1, BDNF, Y27632→ NDM with cAMP, Valpromide, BDNF, Y27632 | growth factor withdrawal | enhanced capase3/7 activity in HD-NSCs | none | Zhang et al. [91] |
| | HD-NSCs and MSNs | TUJ1, GABA, Calbindin, DARPP32 | none | NDM with SHH, DKK1, BDNF, Y27632→ NDM with cAMP, Valpromide, BDNF, Y27632 | growth factor withdrawal | more vulnerable to cell death in HD-NSCs; elevated capase3/7 activity; decreased mitochondrial bioenergetics | none | An et al. [92] |

(Continued.)

**Table 4.** (*Continued.*)

| CAG repeat length | differentiated cell type | markers | differentiation duration | patterning factor | stressor | main finding | compound testing | refs |
|---|---|---|---|---|---|---|---|---|
| 72 | MSNs | DARPP32, GSH-2, DLX2 | none | N2 media with bFGF→N2 media with BDNF | proteasome inhibitor (MG132) | MG132 induced more EM48+ cells<br>Cell engrafted into neonatal brain after 33 weeks post-transplantation | none | Jeon *et al.* [6] |
| **HD iPSC-derived astrocytes** | | | | | | | | |
| 50, 109 | astrocyte | GFAP, s100β | 11–16 weeks | astrocyte medium (ScienCell) | none | time-dependent cytoplasmic vacuolation | none | Juopperi *et al.* [93] |
| 43 | astrocyte | GFAP | 8 weeks | N2B27 media with bFGF→N2B27 medium with ciliary neurotrophic factor | proinflammtory cytokines | cytokine-induced iNOS protected by Xpro1595 (TNF-α inhibitor) | Xpro1595 | Hsiao *et al.* [94] |

NDM, neural differentiation media. NIM, neural induction media.

neurons is rare. So far, only a few studies have indicated that EM48-positive mHtt aggregation can be detected in long-term cultured HD-iPSC-derived neurons with the treatment of the proteasome inhibitor, MG132 [90], or in the HD-NPCs engrafted into neonatal rat brains [6].

The recent advance of genome editing technology provides an excellent opportunity to create isogenic HD-iPSC pairs for better *in vitro* HD modelling via correction of the CAG locus in the *HTT* gene of HD-iPSCs [92,97,98]. An *et al.* [92] were the first to generate genetically corrected isogenic HD-iPSC clones by homologous recombination. The neurons derived from the isogenic line showed rescue of disease phenotypes, including cell death and mitochondrial abnormalities caused by trophic factor withdrawal [92]. Using CRISPR/Cas9 technology, genetically corrected isogenic lines were also generated for *in vitro* neuronal induction. Xu *et al.* [98] used neuronal progenitor cells derived from HD patients, isogenic controls and non-related healthy controls to show that differential gene expression levels caused by genetic background variation were eliminated by using isogenic iPSCs as controls. Among the genes identified in this study, HD-iPSCs carrying $CAG_{180}$ exhibited dysregulation of *CHCHD2*, a genetic risk factor associated with mitochondrial oxidative phosphorylation in late-onset PD [99]. This result suggests that isogenic lines can provide specific controls for studying disease mechanisms and exploring new phenotypes.

Use of cell-based therapy in HD rodent models revealed that motor deficits could be rescued by transplanting human iPSC-derived neural stem cells, but the treatment did not improve striatum function [6,92]. These results indicate that grafted cells were not sufficiently functional as MSNs, or that other types of cells (e.g. striatal interneurons or non-neuronal cells) were required for optimum functional recovery [92]. Glial dysfunction and pathology have been implicated in the pathogenesis of neurodegenerative diseases [94,100–102]. In HD, glial pathology was shown to be associated with striatal neuron dysfunction and other disease phenotypes. In rodent models, striatal transplantation of glia functioned to reduce disease phenotypes, including improving behavioural outcomes, restoring interstitial potassium homeostasis, slowing disease progression and extending lifespan. These results suggest a functional role of glia in HD, implying the potentials for iPSC-based glial replacement therapies [102]. Thus, we provide a summary of protocols for iPSC differentiation into HD-relevant cells, including MSNs and astrocytes (table 4).

Several studies have used iPSC-derived neurons to investigate HD-associated pathogenesis and identify effective therapeutic strategies or chemical compounds. DNA damage, including damage response and repair machinery, was shown to be involved in HD pathogenesis. ATM-p53 signalling was enhanced by phosphorylation of p53 and $H_2AX$ in HD-iPSC-derived neurons. Accordingly, neocarzinostatin treatment improved cell viability through activating DNA damage repair pathways [103]. Chiu *et al.* [89] showed activation of the $A_2AR$-PKA pathway protected HD-iPSC-derived MSNs from oxidative stress-induced DNA damage and cell death. In addition, microglia exhibited proinflammatory gene expression in HD [104], suggesting inflammatory processes may be associated with HD pathology. In co-cultured MSNs and astrocytes, both derived from HD-iPSCs [93,94], MSN death was reduced after treatment with a TNF-α inhibitor, Xpro1595. This result suggests that reducing cytokine-induced iNOS expression in astrocytes can reduce HD phenotypes [94]. Store-operated channel (SOC) was associated with calcium influx in HD-iPSC-derived MSNs, and an NF-κB inhibitor effectively decreased SOC-mediated calcium entry [90]. Moreover, inhibitors in the forms of microRNAs and peptides have been explored as therapeutic strategies for HD. The miR196a reduced mHtt aggregation through mediating the ubiquitin-proteasome pathway [105], while P110-TAT, a peptide inhibitor of dynamin-related protein (Drp1), reduced HD phenotypes, including excessive mitochondrial fission and cell death [106].

# 5. Modelling neurodegenerative diseases with three-dimensional brain organoids

2D culture systems have been widely used as human cell-based platforms for modelling neurodegenerative disease, and these models have been useful for the discovery of potential treatments. However, 2D culture systems are not suitable to mimic the intricately structured *in vivo* environment. Therefore, 3D culture systems have become valuable tissue models, which include extracellular matrix and cell–cell interactions that are necessary for proper differentiation, proliferation and cell-based functions [107,108]. Hence, human iPSCs grown in a 3D culture system represents a straightforward substrate for fundamental studies on the pathophysiology of human neurodegenerative disease.

PSCs have the capacity for self-organization and can develop into 3D structures resembling mini-organs, including the cerebral cortex. ESCs and iPSCs both exhibit the capability of developing through self-regulated processes into cortical neuroepithelial structures with multilayered neuroepithelium that resembles the progenitor zones of the embryonic cortex [109,110]. Therefore, this culture system is rapidly becoming an important platform for modelling human cortical development and neurodegenerative disease. However, these self-organizing cerebral spheres still lack essential developmental and patterning cues to develop into fully formed and mature organs. Thus, strategies are under development to identify and provide appropriate physico-chemical cues for transforming brain organoids into the same tissue patterns as occur *in vivo*. For example, cells have been embedded in scaffolding materials such as matrigel, self-assembling peptide (SAP) matrix and PNIPAAm-PEG hydrogel [15,86,87,111–119]. To increase oxygen uptake, culture conditions and vessels have been adjusted, including raising the oxygen percentage [116,120], use of a culture shaker, and growth of cultures in a spinning bioreactor [15,86,87,112,113,115,121]. Therefore, a growing number of protocols has been developed for generating cortex-like tissue *in vitro* (table 5).

Human iPSCs showed the capability of developing into a dorsal telencephalon-like structure through self-organized 3D cultures [114,124]. Lancaster *et al.* [15] used a spinning bioreactor to improve diffusion of oxygen and nutrient supply to the spheroids, meanwhile promoting non-neural cell production without neuroectoderm formation by blocking TGF-ß and BMP pathways. These cerebral cortex organoids recapitulated features of human cortical development including characteristic progenitor zone organization with abundant outer radial glial stem cells. Pasca *et al.* [122]

**Table 5.** Human iPSC-derived 3D neural models.

| brain region in organoid | starting cells | patterning factor | extracellular scaffold | device/ bioreactor | days | main finding | refs |
|---|---|---|---|---|---|---|---|
| cerebral cortex organoids | hiPSC-derived NSCs | DKK1, BMPRIA-Fc, SB431542 | Matrigel | none | approximately 50 days | polarized radial glia, intermediate progenitor, and spectrum of layer-specific cortical neurons | Mariani *et al.* [114] |
| cerebral organoids | ESC/hiPSC | insulin, vitamin A, Retinoic acid | Matrigel | spinning bioreactor | 2 months (survived for 10 months) | well-organized progenitor zone surrounded with outer radial glial stem cells | Lancaster *et al.* [15] |
| cerebral cortex organoids | hiPSCs | FGF2, EGF, BDNF, NT3 | none | none | 180 days | laminated cerebral cortex-like structure contain deep and superficial cortical layers and functional neurons | Pasca *et al.* [122] |
| forebrain, midbrain and hypothalamic organoids | hiPSCs | **forebrain organoids**: AA, BDNF, GDNF, TFG-*β*, cAMP<br><br>**midbrain organoids**: SHH, LDN-193189, CHIR99021, PUR, FGF-8, BDNF, GDNF, AA, TFG-b, cAMP<br><br>**hypothalamic organoids**: Wnt-3A, SHH, PUR, FGF-2, CTNF | none | mini-spinning bioreactor | **forebrain**: >80 days<br><br>**midbrain**: 75 days<br><br>**hypothalamus**: 40 days | **forebrain organoids**: progenitor zone, neurogenesis, human specific outer radial glia layer<br><br>**midbrain organoids**: TH⁺ neurons co-expressing dopaminergic neuron markers (FOXG2, DAT, NURR1, PITX3)<br><br>**hypothalamic organoids**: peptidergic neuronal markers (POMC, VIP, OXT, NPY, OTP) | Qian *et al.* [121] |
| forebrain organoids | hiPSCs | cAMP, insulin, BDNF, GDNF | BME matrix | rocking cell culture shaker | 35 days | forebrain-like structure: ventricular zone, inner and outer subventricular zone, cortical plate-like area | Krefft *et al.* [119] |
| midbrain organoids | hiPSC-derived NSCs | BDNF, GDNF, db-cAMP, AA, TFG-*β*, PUR | BD Matrigel | orbital shaker | approximately 60 days | structure organized, dopaminergic neurons, astroglial (4%) and oligodendrocyte (29.6%) | Monzel *et al.* [86] |
| midbrain organoids | hPSCs | SB431542, Noggin, CHIR99021; SHH, FGF8; BDNF, GDNF, AA, db-cAMP | BD Matrigel | orbital shaker | 65–84 days | functional dopaminergic neurons<br><br>substantia nigra-like tissue (neuromelanin-like granules) | Jo *et al.* [123] |

(*Continued.*)

**Table 5.** (Continued.)

| brain region in organoid | starting cells | patterning factor | extracellular scaffold | device/ bioreactor | days | main finding | refs |
|---|---|---|---|---|---|---|---|
| 3D nigrostriatal dopaminergic neurons | hiPSC-derived NSCs | AA, BDNF, GDNF, TGF-$\beta$3, cAMP, PMA, | BD Matrigel | microfluidic bioreactor | 30 days | electrophysiologically active dopaminergic neurons; 19% of tyrosine hroxylase-positive neurons | Moreno et al. [115] |
| 3D striatal neurons | hESC/hiPSC-derived striatal NSCs | DKK-1, PUR, BDNF, GDNF, cAMP, IGF-1 | PNIPAAm-PEG hydrogel | none | 60 days | functional striatal neurons; cell population diversity: 43% DARPP32 neurons and 27% glial; Transplantation 3D-derived striatal progenotor into HD mice improved motor coordination and increased survival | Adil et al. [111] |

PUR, purmophamine; AA, ascorbic acid; PMA, phorbol 12-myristate 13-acetate.

expanded self-organized cortical spheroids that contained astrocytes surrounding electrophysiologically matured and functional neuronal synapses. Furthermore, defined culture conditions were used to generate forebrain, midbrain and hypothalamic organoids in a spinning bioreactor. Human iPSC-derived brain organoids recapitulated human-specific outer radial glia cell layers in cortical development and other key features, including progenitor zone organization, neurogenesis and gene expression [121]. Moreover, Jo et al. [123] generated 3D ventral midbrain-like organoids that contain functional midbrain dopaminergic neurons and neuromelanin granule, similar to human substantia nigra tissue. Thus, 3D neural organoids faithfully mimic the basic processes of brain development and functional patterning of brain regions, providing a highly advantageous model for studying human neurodegenerative diseases or other neurological disorders [34].

It is clear that iPSC-derived spheroids or organoids could recapitulate neurodevelopmental processes, potentially allowing the investigation of developmental disorders related to the human cerebral cortex (table 6). Recently, patient-derived iPSCs carrying a *CDK5RAP2* mutation were used by Lancaster et al. to successfully model primary microcephaly in 3D cerebral organoids; the cultures recapitulated premature neural differentiation and decreased numbers of radial glial cells, producing overall smaller organoids than wild-type controls [15]. This study provided a striking example of modelling neurodevelopmental disorders in human cell culture systems, which have not been successfully recapitulated in mouse models. Miller–Dieker syndrome (MDS), a brain malformation (lissencephaly) syndrome, is a contiguous gene deletion of chromosome 17p13.3 involving the *LIS1* and/or *YWHAE* genes (coding for 14.3.3$\varepsilon$) [127]. MDS-iPSC-derived organoids mimicked brain malformations with a smaller size, which was likely caused by premature neurogenesis and alterations of cortical niche architecture. Based on this model system, the authors further discovered that the LIS1/NDEL1/14.3.3$\varepsilon$ complex was associated with cortical niche architecture and the deletion leads to non-cell-autonomous disturbance of $\beta$-catenin signalling [113]. Models of idiopathic autism spectrum disorder (ASD) based on patient-derived iPSCs revealed that the ASD organoids recapitulated human first-trimester telencephalic development, but with over-production of neuronal progenitors due to a shorter cell cycle length. Also, the production of GABAergic neurons was induced due to increased expression of FOXG1, a transcription factor involved in early cortical neuron production and associated with prenatal microcephaly of some ASDs subtype [125]. Furthermore, Srikanth et al. [117] modelled a neuropsychiatric disease using DISC1-disrupted iPSCs to generate cerebral organoids. DISC1-mutant cerebral organoids display disorganized structural morphology and impaired proliferation. The authors further reported that DISC1 isoforms were associated with elevation of baseline WNT signalling in NPCs, which resulted in morphological and neurodevelopmental consequences through alterations in cell fate and progenitor migration. Recently, Zika virus was linked to infants born with microcephaly. Human brain organoids showed Zika virus affected neural stem cells, and pure forebrain organoids showed Zika virus affected cell proliferation in the ventricular zone. These findings supported the notion that the pathogen directly affects fetal brain development processes and

**Table 6.** Modelling neurological disorder with 3D brain organoids derived from disease iPSCs.

| 3D brain-like tissue | model disease | gene mutation | starting cells | patterning factor | extracellular scaffold | bioreactor/ oxygen | differentiation period | main finding | refs |
|---|---|---|---|---|---|---|---|---|---|
| **neurodevelopmental disorder** | | | | | | | | | |
| forebrain organoids | sever microcephaly (MCPH) | *CDK5RAP2* | MCPH-iPSCs | AA, BDNF, GDNF, TGF-β, cAMP | none | mini-spinning bioreactor | 2 months | MCPH-forebrain organoids: premature neural differentiation (only occasional neuroepithelial region) decreased numbers of radial glia | Lancaster *et al.* [15] |
| telencephalic organoids | autism spectrum disorder (ASD) | idiopathic ASD | ASD-iPSC-derived NSCs | DKK1 BMPRIA-Fc, SB431542 | Matrigel | ultralow-attachment plates | 4–5 weeks | ASD-cortical organoids reflect human midfetal telencephalic development accelerated cell cycle; Overproduction GABAergic inhibitory neurons cause by increased FOXG1 | Mariani *et al.* [125] |
| forebrain organoids | Miller–Dieker syndrome (MDS) | *LIS1, YWHAE* | MDS-iPSCs | A83, LDN, IWR-1, CHIR99021 | BME matrix | rocking cell culture shaker | 35 days | MDS-forebrain organoids: reduced expansion rate caused by premature neurogenesis cortical niche alterations architecture leading to a non-cell-autonomous disturbance of β-catenin signalling | Iefremova *et al.* [113] |
| cerebral organoids | neuro-psychiatric disease | *DISC1* | DISC1-disrupted-iPSC | XAN939, CHIR99021 | Matrigel droplets | ultralow-attachment plates | 19 days | DISC1-disrupted cerebral organoids: disorganized structure morphology and impaired proliferation rescued with a WNT antagonist | Srikanth *et al.* [117] |

**Table 6.** (*Continued.*)

| 3D brain-like tissue | model disease | gene mutation | starting cells | patterning factor | extracellular scaffold | bioreactor/ oxygen | differentiation period | main finding | refs |
|---|---|---|---|---|---|---|---|---|---|
| **neurodegenerative disease** | | | | | | | | | |
| hiPSC-derived 3D neural tissue | AD | Aβ oligomer added | iPSC-derived NSCs | SHH, FGF8, NGF, BMP9 | SAP matrix | poly-L-ornithine-Laminin coated dish | | p21-activated kinase mediated Aβ oligomers sensing and connected to AD processes | Zhang et al. [118] |
| AD-iPSC-derived 3D neuro-spheroids | AD | sporadic AD | AD-iPSC-derived NSCs | dorsomorphin, SB431542, FGF2, EGF, BDNF, NT3 | none | ultralow-attachment plates | 9 weeks | BACE1 or γ-secretase inhibitors showed less effects on Aβ production in 3D neuro-spheroids than 2D neurons | Lee et al. [126] |
| AD-iPSC-derived 3D neural tissue | AD | APP duplication, PSEN1 (M146 L), PSEN1 (A246E) | AD-iPSCs | A83, LDN, bFGF, EGF, BDNF, NT3 | Matrigel | non-adherent Petri-dish; 40% O2 | 90 days | recapitulate AD-like pathologies (Aβ aggregation, hyperphosphorylated tau protein, endosome abnormalities) reduction of amyloid and tau pathology treated with β- and γ-secretase inhibitors | Raja et al. [116] |
| fPD-iPSC-derived 3D neurospheroids | fPD | LRRK2 | fPD-iPSC | bFGF, EGF, LIF | none | ultralow-attachment plates | | changes of gene expression in synaptic transmission, Toll-like receptor signalling pathways and neurotransmitter level regulation | Son et al. [87] |
| HD-iPSC-derived cortical organoids | HD | mHtt | HD-iPSC | retinoic acid | Matrigel droplets | orbital shaker | 105 days | HD organoids: abnormality in neural rosette formation disrupted cytoarchitecture in cortical organoids the impairment can be rescued by molecular and pharmacological approaches | Conforti et al. [112] |

AA, ascorbic acid; LDN, LDN193189; SAP matrix, self-assembling peptide matrix.

royalsocietypublishing.org/journal/rsob   *Open Biol.* **9**: 180177

suggested that Zika virus may be involved in apoptosis of neuron progenitors [128,129].

Brain organoids show faithful recapitulation of brain development and organization of functional cells. These highly complex and dynamic 3D networks among neurons and glia also provide a means to further understand neurodegenerative disorders from a more systemic view. As previously noted, additional cellular insults are often required to generate disease-relevant phenotypes while modelling with iPSC-derived neurons. Currently, explorations into the modelling of neurodegenerative disorder phenotypes using 3D organoids for therapeutic development are becoming increasingly prevalent.

For recapitulating AD pathology, 3D culture systems were shown to exhibit extracellular deposition of A$\beta$ and increased pTau level [116,118,126,130,131]. Using genetically modified immortalized hNPCs with familial AD mutations (*APP* and *PSEN1*), the aforementioned phenotypes were shown along with higher expression of four-repeat adult Tau (4R Tau) isoforms [130,131]. Zhang *et al.* [118] used a 3D hydrogel-based culture system to shorten the time necessary of generating functional neurons and applied treatments to diminish A$\beta$ oligomer production in AD organoids. In addition, 3D organoids have been used to reveal new disease phenotypes as well as AD pathology. For example, Raja *et al.* [116] identified endosome abnormalities associated with different mutations in *APP* (duplication) or *PSEN1* (M146 L, A246E) by using AD-iPSC-derived brain organoids. Finally, using iPSCs derived from sporadic AD patients, Lee *et al.* [126] recapitulated both A$\beta$ and pTau pathology with NPC-derived 3D neurospheroids.

3D organoids also exhibit certain phenotypes, which are not observed in 2D culture systems. In HD-iPSC-derived brain organoids, the length of CAG repeat was associated with neural differentiation capacity [112]. The HD-iPSCs carrying longer CAG repeats (Q109 and Q180) exhibited complete failure of neuroectodermal acquisition; however, shorter CAG repeats (Q60) showed milder abnormalities in neural rosette formation and disrupted cytoarchitecture in cortical organoids. These findings were not observed in 2D culture systems and suggest that 3D culture conditions can accelerate neuronal maturation and recapitulate disease pathogenesis. From this handful of studies using a 3D culture system for modelling neurological diseases, one can easily see that this platform holds immense opportunities and the potential for studying human-specific neurological diseases. However, it should be borne in mind that certain limitations will continue to hamper the study of all neuropathologies, and especially adult-onset diseases.

# 6. The applications and challenges of iPSC-derived three-dimensional cultures

The self-organization of brain organoids without embryonic surroundings allows researchers to inspect neurodevelopmental process. Current models have clearly demonstrated that exogenous cues are essential for organoids to develop into well-patterned mature brains. Also, variability in quality and brain regions are important issues for disease modelling and drug testing. For modelling of adult-onset diseases, immature structures or neuron-only organoids may not be sufficient to reflect complex neurobiological processes and drug effects. These factors will impact test results from the laboratory to the clinic. Therefore, robust and mature 3D organ culture models will help to accelerate neurological research and neuropharmaceutical development.

3D neuronal organoids allow penetration of small molecules and sufficient oxygen and nutrient supply that are essential for the survival of the innermost cells. Thus, the 3D-iPSC model enables researchers to analyse molecular and pharmacological effects in a complex tissue system. Taking AD as an example, the different culture systems show different responses to test compounds. For example, in an AD patient iPSC-derived 3D culture system, amyloid and Tau pathology were significantly reduced after treatment with $\beta$- and $\gamma$-secretase inhibitors [116,126]; however, Lee *et al.* [126] indicated that $\beta$- and $\gamma$-secretase inhibitors work more effectively in decreasing A$\beta$ levels in AD-NPCs-derived 2D neurons than in 3D spheroids [132]. Additionally, it was shown that p21-activated kinase was associated with A$\beta$ oligomer-mediated AD pathogenesis in the 3D organoids, which was not observed in 2D culture-derived neurons [118]. Potentially, drug responses and pathogenesis may vary according to the cell population and environment that is present, suggesting the 3D culture systems may more reliably reflect the conditions in patient brains. Moreover, the pathogenesis of adult-onset neurodegenerative diseases is generally considered to result from long-term chronic exposure to neurotoxicants. The cells in 3D-cultured organoids are viable for much longer times than 2D-cultured neurons, as long as nutrients and oxygen are efficiently supplied. Therefore, the 3D culture system provides an exciting platform for exploring pathogenesis caused by long-term neurotoxicant exposure and chronic cellular response. Smirnova *et al.* [133] administered neurotoxicants (MPP$^+$ and rotenone) to dopaminergic neurons in 3D neuronal models derived from PD-iPSCs, and reported that dopaminergic neurons responded to toxicant exposure by upregulating one-carbon metabolism, transsulfuration pathways (ASS1, CTH and SHTM2) and $\alpha$-synuclein associated microRNAs. This report provided a pioneering example of using a human cell model for investigating neurotoxicology and sporadic PD [133]. 3D-iPSC models provide a more realistic platform to investigate neurodevelopment and neurological disorders, providing an excellent multifaceted system with which to investigate chronic effects of drug treatment.

Despite the great potential of 3D organoids, low numbers of non-neuronal cells and lack of vascularization may limit their utility. The interaction between non-neuronal cell populations and neuronal function has become one of the most important features that can be modelled by 3D cerebral organoids. For example, the neuronal function was characterized by synaptic connections and electrophysiological activity in midbrain organoids, which contained organized dopaminergic neurons, astroglia (4%) and oligodendrocytes (29.6%) [86]. Likewise, striatal neurons fired action potentials in Hydrogel-embedded 3D culture systems that contained 27% glia [86,111]. In a mouse model of HD, motor coordination and lifespan were improved after transplantation with 3D striatal progenitors [111]. Pasca *et al.* also indicated that neuronal synapse function was improved when surrounded by astrocytes (approx. 20%) in cortical spheroids [122]. The complexity of the extracellular environment in 3D culture systems presents a relatively realistic platform for studying neurobiology and neurological diseases. Additionally, the vasculature is not present in the early developing neocortex ahead of blood vessel invasion; however, vascularization is essential for neuronal progenitor differentiation in the subventricular zone

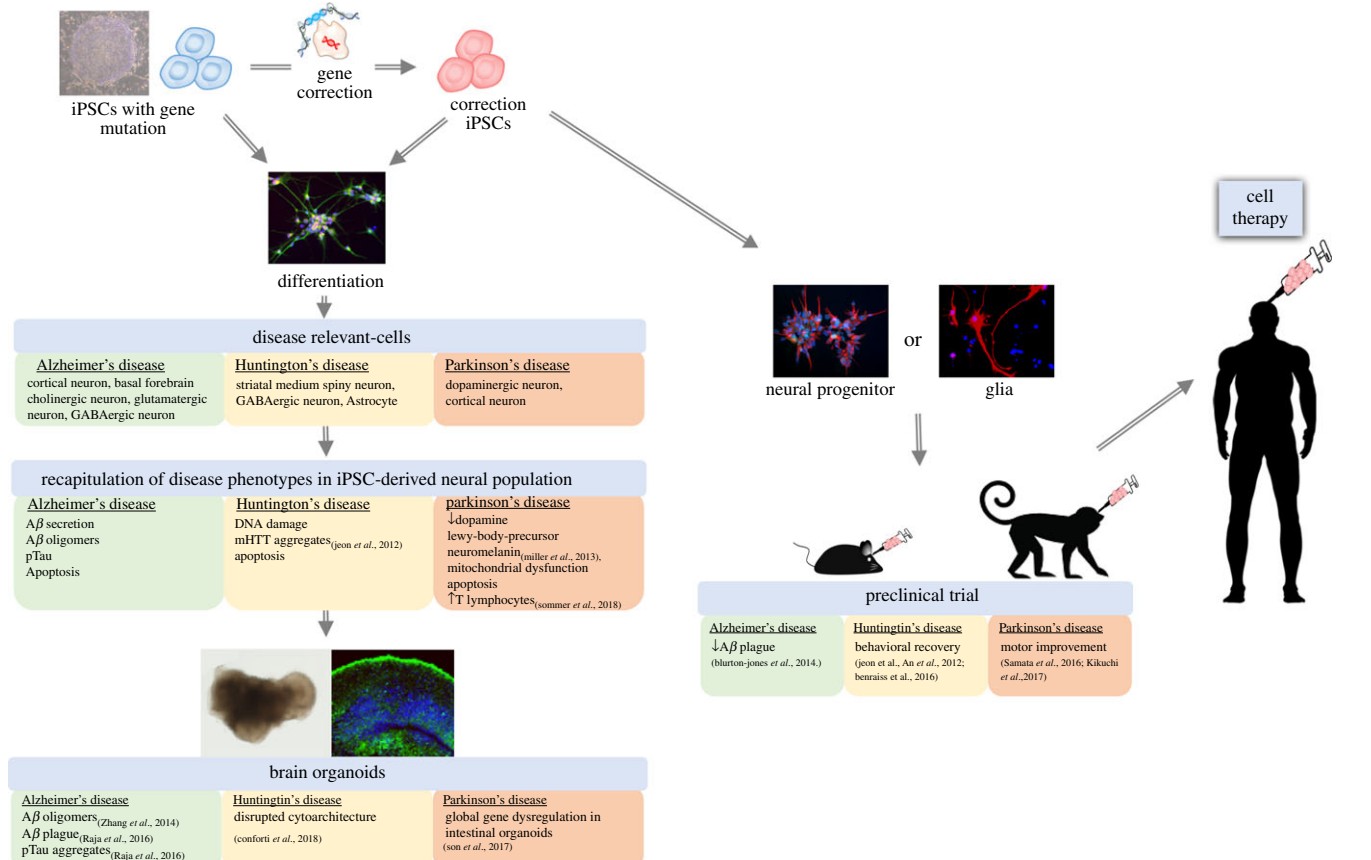

**Figure 1.** Applications of human induced pluripotent stem cells in neurodegenerative disease.

during late development [134]. This limitation also hampers PD modelling because degeneration of nigrostriatal projection neurons is the primary cause of PD symptoms, and vascularization would be an essential first step to begin the enormous task of rebuilding a functional nigrostriatal circuit in a culture setting. Strategies such as building a vascular microenvironment by microfluidic chambers have been developed to address the lack of vascularization, further mimicking the physiological niche for neurogenesis [135–137]. In the tissue scaffold, pre-capillary networks were built by co-culture of pericytes and early vascular cells derived from hPSCs [138,139]. This approach was able to facilitate the generation of physiologically relevant vascular networks for neurogenesis. Thus, perfusion-based human iPSC-derived 3D brain organoid platforms represent increasingly realistic *in vitro* models for neurodegenerative diseases.

## 7. Conclusion

Accumulating evidence demonstrates that human iPSCs can be used as a reliable basis for the generation of disease-relevant cell types to explore the mechanisms underlying human disease. Through continuous propagation and efficient differentiation of patient-derived iPSCs into specific neuronal subtypes and 3D organoids, we are now able to better recapitulate the cellular progression toward neurodegeneration *in vitro*. This capability allows researchers to discover new methods to manipulate iPSC-derived neurons, uncovering novel disease phenotypes *in vitro* in the hope of eventually identifying novel clinical interventions. For example, age and disease-associated phenotypes can be recapitulated by progerin impairment, and this accelerated ageing may be used

in combination with PD-iPSC-derived neurons [33]. The shortening of telomeres is a typical feature of ageing, which can be induced by overexpression of progerin in neurons, helping to drive the development of PD phenotypes [33]. This discovery suggests that ageing-accelerated disease modelling may be applicable in several neurodegenerative diseases, including AD [140], PD [141] and HD [83]. Since impairment of nuclear pore structures has been identified in all of the aforementioned diseases, researchers should be aware of which cellular consequences stem from progerin overexpression and which are specifically related to the disease. In addition, region-specific brain organoids provide a platform to investigate early-stage phenotypes, which may potentially serve as biomarkers for early diagnosis and drug targets for preventing disease progression. A key to fulfilling the promise of iPSC technology to discover new medicines for the human neurodegenerative disease will be to incorporate the findings from iPSCs with other successful models. Since brain organoids lack vascularization, patterning cues and complex cell-cell interactions, such as those found in the nigrostriatal pathway [142], animal models will continue to provide essential readouts for drug evaluation, especially those related to physiological interactions and disease-associated behavioural phenotypes (e.g. cognitive impairment in AD and motor defects in PD and HD). We summarize current progress and applications of human iPSCs in neurodegenerative diseases including AD, PD and HD in figure 1. Although there are still many obstacles to overcome before iPSC-based technology can be used directly in clinical applications, the combination of iPSC technology with genome editing, organoid engineering and other technologies will certainly accelerate the development of new medicines for human neurodegenerative disease and eventually may cure these devastating diseases by cell replacement therapy.

Data accessibility. This article has no additional data.
Competing interests. We declare we have no competing interests.

Funding. This work was supported by grants from Academia Sinica (Thematic Project, AS-106-TP-B13) and the Ministry of Science and Technology (MOST107-2321-B-010-007).

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
