## [Reviewer comments · Open Biology]

Review History

RSOB-18-0177.R0 (Original submission)

Review form: Reviewer 1

Recommendation

Accept with minor revision (please list in comments)

Are each of the following suitable for general readers?

- a) **Title**
Yes

- b) **Summary**
Yes

c) Introduction

Yes

Is the length of the paper justified?

Yes

Should the paper be seen by a specialist statistical reviewer?

No

Is it clear how to make all supporting data available?

Not Applicable

Is the supplementary material necessary; and if so is it adequate and clear?

Not Applicable

Do you have any ethical concerns with this paper?

No

Comments to the Author

This is a timely, interesting and rather comprehensive review on the use of human iPSCs in modeling neurodegenerative diseases. 3 main neurodegenerative diseases were discussed, i.e. AD, PD and HD. Overall, the write up is informative and the tables provided that summarize the various iPSC-linked disease models generated thus far by different groups serve as a good resource.

Notwithstanding the above, I have some comments/suggestions that might help to improve the article further, as discussed below:

1. Although current animal models of neurodegenerative diseases are well recognized for their limitations, they have nonetheless shed important insights into the pathogenesis of various neurodegenerative disorders. Importantly, they provide the associated behavioral phenotype (e.g. cognitive or motor impairments) that could be quantified, which also serve as useful readouts for drug evaluation. This is something that 2D and even 3D iPSC-based disease models cannot recapitulate. The authors may wish to take this into consideration in their discussion.
2. The maturity of iPSC-derived neurons and the time for them to age sufficiently to manifest the disease-associated phenotypes is indeed an issue (p. 4). The authors stated that the issue is addressed by "the use of multiple well-characterized iPSC lines and isogenic controls". Arguably, these lines may resolve the variability issue but not the time issue.
3. Related to the above, for age-related neurodegenerative diseases like AD and PD, substantial length of time is needed for the phenotype to manifest. Hence, the use of progerin-induced aging as a strategy to accelerate the aging process, which the authors have also highlighted. However, is progerin-induced aging an appropriate model for AD and PD? This is something that merits discussion.
4. The section "Modeling neurodegenerative diseases with by 3D brain model" is grammatically incorrect (i.e. with or by?). Importantly, the reported generation and characterization of 3D ventral midbrain organoid model that is relevant to PD (e.g. Jo et al., 2017 Cell Stem Cell) was not discussed.
5. The authors have discussed several limitations surrounding current 3D organoid models. One major limitation with the ventral midbrain model is its primitive structure, which in no way

recapitulate the nigrostriatal pathway that degenerates in human PD brain. The authors might want to include this point in their revised discussion.

6. Some missing references: "...Mao-B inhibitors or overexpression...for PD treatment" (p. 10); "neuronal function was characterized...oligodendrocytes (29.6%)" (p. 17)

7. Parkin R275W is incorrectly stated as PINK1 R275W (p. 9)

8. The manuscript is peppered with typographic errors. For example, "PRAK7" instead of "PARK 7" (p. 8); "glucocerebrocidas" instead of "glucocerebrocidase" (p. 10)

Decision letter (RSOB-18-0177.R0)

05-Nov-2018

Dear Dr Wu

We are pleased to inform you that your manuscript RSOB-18-0177 entitled "Opportunities and challenges for the use of induced pluripotent stem cells in modeling neurodegenerative disease" has been accepted by the Editor for publication in *Open Biology*. The reviewer has recommended publication, but also suggest some minor revisions to your manuscript. Therefore, we invite you to respond to the reviewer's comments and revise your manuscript.

Please submit the revised version of your manuscript within 14 days. If you do not think you will be able to meet this date please let us know immediately and we can extend this deadline for you.

1) A text file of the manuscript (doc, txt, rtf or tex), including the references, tables (including captions) and figure captions. Please remove any tracked changes from the text before submission. PDF files are not an accepted format for the "Main Document".

2) A separate electronic file of each figure (tiff, EPS or print-quality PDF preferred). The format

should be produced directly from original creation package, or original software format. Please note that PowerPoint files are not accepted.

3) Electronic supplementary material: this should be contained in a separate file from the main text and meet our ESM criteria (see <http://royalsocietypublishing.org/instructions-authors#question5>). All supplementary materials accompanying an accepted article will be treated as in their final form. They will be published alongside the paper on the journal website and posted on the online figshare repository. Files on figshare will be made available approximately one week before the accompanying article so that the supplementary material can be attributed a unique DOI.

Online supplementary material will also carry the title and description provided during submission, so please ensure these are accurate and informative. Note that the Royal Society will not edit or typeset supplementary material and it will be hosted as provided. Please ensure that the supplementary material includes the paper details (authors, title, journal name, article DOI). Your article DOI will be 10.1098/rsob.2016[last 4 digits of e.g. 10.1098/rsob.20160049].

4) A media summary: a short non-technical summary (up to 100 words) of the key findings/importance of your manuscript. Please try to write in simple English, avoid jargon, explain the importance of the topic, outline the main implications and describe why this topic is newsworthy.

Images

Data-Sharing

It is a condition of publication that data supporting your paper are made available. Data should be made available either in the electronic supplementary material or through an appropriate repository. Details of how to access data should be included in your paper. Please see <http://royalsocietypublishing.org/site/authors/policy.xhtml#question6> for more details.

Data accessibility section

Sincerely,

The Open Biology Team
<mailto:openbiology@royalsociety.org>

Reviewer(s)' Comments to Author:

Referee:

Comments to the Author(s)

This is a timely, interesting and rather comprehensive review on the use of human iPSCs in modeling neurodegenerative diseases. 3 main neurodegenerative diseases were discussed, i.e. AD, PD and HD. Overall, the write up is informative and the tables provided that summarize the various iPSC-linked disease models generated thus far by different groups serve as a good resource.

Notwithstanding the above, I have some comments/suggestions that might help to improve the article further, as discussed below:

1. Although current animal models of neurodegenerative diseases are well recognized for their limitations, they have nonetheless shed important insights into the pathogenesis of various neurodegenerative disorders. Importantly, they provide the associated behavioral phenotype (e.g. cognitive or motor impairments) that could be quantified, which also serve as useful readouts for drug evaluation. This is something that 2D and even 3D iPSC-based disease models cannot recapitulate. The authors may wish to take this into consideration in their discussion.
2. The maturity of iPSC-derived neurons and the time for them to age sufficiently to manifest the disease-associated phenotypes is indeed an issue (p. 4). The authors stated that the issue is addressed by "the use of multiple well-characterized iPSC lines and isogenic controls". Arguably, these lines may resolve the variability issue but not the time issue.
3. Related to the above, for age-related neurodegenerative diseases like AD and PD, substantial length of time is needed for the phenotype to manifest. Hence, the use of progerin-induced aging as a strategy to accelerate the aging process, which the authors have also highlighted. However, is progerin-induced aging an appropriate model for AD and PD? This is something that merits discussion.
4. The section "Modeling neurodegenerative diseases with by 3D brain model" is grammatically incorrect (i.e. with or by?). Importantly, the reported generation and characterization of 3D ventral midbrain organoid model that is relevant to PD (e.g. Jo et al., 2017 Cell Stem Cell) was not discussed.
5. The authors have discussed several limitations surrounding current 3D organoid models. One major limitation with the ventral midbrain model is its primitive structure, which in no way recapitulate the nigrostriatal pathway that degenerates in human PD brain. The authors might want to include this point in their revised discussion.
6. Some missing references: "...Mao-B inhibitors or overexpression...for PD treatment" (p. 10); "neuronal function was characterized...oligodendrocytes (29.6%)" (p. 17)
7. Parkin R275W is incorrectly stated as PINK1 R275W (p. 9)
8. The manuscript is peppered with typographic errors. For example, "PRAK7" instead of "PARK 7" (p. 8); "glucocerebrocidase" instead of "glucocerebrocidase" (p. 10)

Author's Response to Decision Letter for (RSOB-18-0177.R0)

See Appendix A.

Decision letter (RSOB-18-0177.R1)

03-Dec-2018

Dear Dr Wu

We are pleased to inform you that your manuscript entitled "Opportunities and challenges for the use of induced pluripotent stem cells in modeling neurodegenerative disease" has been accepted by the Editor for publication in Open Biology.

Sincerely,

The Open Biology Team
mailto:openbiology@royalsociety.org

Appendix A

MS ID#: RSOB-18-0177

Current Title: Opportunities and challenges for the use of induced pluripotent stem cells in modeling neurodegenerative disease

Reviewer's Comments to Author:

Formatted: Font color: Text 1, Not Highlight

Formatted: Font color: Text 1

Referee:

Comments to the Author(s)

This is a timely, interesting and rather comprehensive review on the use of human iPSCs in modeling neurodegenerative diseases. 3 main neurodegenerative diseases were discussed, i.e. AD, PD and HD. Overall, the write up is informative and the tables provided that summarize the various iPSC-linked disease models generated thus far by different groups serve as a good resource.

Notwithstanding the above, I have some comments/suggestions that might help to improve the article further, as discussed below:

1. Although current animal models of neurodegenerative diseases are well recognized for their limitations, they have nonetheless shed important insights into the pathogenesis of various neurodegenerative disorders. Importantly, they provide the associated behavioral phenotype (e.g. cognitive or motor impairments) that could be quantified, which also serve as useful readouts for drug evaluation. This is something that 2D and even 3D iPSC-based disease models cannot recapitulate. The authors may wish to take this into consideration in their discussion.

Response:

We thank reviewers for this and other thoughtful suggestions; we believe that addressing each one has greatly improved the quality of our manuscript. We agree that animal models provide important insights into the pathogenesis of neurodegenerative diseases, and it is inarguable that *in vitro* iPSC-based models cannot recapitulate the behavioral phenotypes, such as motor dysfunction in PD and impaired cognition in AD. Thus, we have revised the Conclusion accordingly, and we now suggest the use of iPSC-based models in combination with animal models for drug development. In this way, novel interventions can be evaluated in these complementary systems prior to clinical application (Page 18, Line 579).

Formatted: Font color: Text 1

Formatted: Font: Italic, Font color: Text 1

Formatted: Font color: Text 1

Formatted: Font color: Text 1

2. The maturity of iPSC-derived neurons and the time for them to age sufficiently to manifest the disease-associated phenotypes is indeed an issue (p. 4). The authors stated that the issue is addressed by "the use of multiple well-characterized iPSC lines and isogenic controls". Arguably, these lines may resolve the variability issue but not the time issue.

Response:

We revised the Introduction accordingly, mentioning that some issues of timing could be potentially addressed by multiple treatments or long-term 3D organoid cultures. For most diseases of aging, multiple treatments are required to promote the expression of disease-associated phenotypes in cellular models. This approach may also be improved by utilizing long-term 3D organoid cultures. These complex structures

uniquely provide a human organ-like tissue that is amenable to long-term culturing for disease modeling (Page 3, Line 90).

Formatted: Font color: Text 1

Formatted: Font color: Text 1

3. Related to the above, for age-related neurodegenerative diseases like AD and PD, substantial length of time is needed for the phenotype to manifest. Hence, the use of progerin-induced aging as a strategy to accelerate the aging process, which the authors have also highlighted. However, is progerin-induced aging an appropriate model for AD and PD? This is something that merits discussion.

Response:

Progerin overexpression causes telomere shortening, which is a typical feature of aging. Thus, we revised the Conclusion and now discuss the potential approach of inducing premature aging with progerin in combination with iPSCs derived from other diseases, including HD and AD. However, the cellular pathology that results from progerin overexpression should be excluded in this type of experiment to distinguish disease-related consequences (Page 18, Line 567).

Formatted: Font color: Text 1, Not Highlight

Formatted: Font color: Text 1

Formatted: Font color: Text 1, Not Highlight

Formatted: Font color: Text 1

Formatted: Font color: Text 1, Not Highlight

Formatted: Font color: Text 1

Formatted: Font color: Text 1

4. The section “Modeling neurodegenerative diseases with by 3D brain model” is grammatically incorrect (i.e. with or by?). Importantly, the reported generation and characterization of 3D ventral midbrain organoid model that is relevant to PD (e.g. Jo et al., 2017 Cell Stem Cell) was not discussed.

Response:

We thank reviewer for pointing out the error and have revised the subtitle accordingly. We also have added discussion of pluripotent stem cell-derived 3D ventral midbrain organoids to model PD (e.g. Jo et al., 2017 Cell Stem Cell). This midbrain-like organoid recapitulated the dopamine production, as well as human-specific neuromelanin-containing cells, suggesting that the 3D organoid model may provide an excellent means to gain further insights into the mechanisms underlying PD and develop new drugs for PD treatment (Page 14, Line 429).

Formatted: Normal, Left

Formatted: Font: Times New Roman, 12 pt, Font color: Text 1

Formatted: Font color: Text 1

Formatted: Font color: Text 1

Formatted: Line spacing: 1.5 lines

5. The authors have discussed several limitations surrounding current 3D organoid models. One major limitation with the ventral midbrain model is its primitive structure, which in no way recapitulate the nigrostriatal pathway that degenerates in human PD brain. The authors might want to include this point in their revised discussion.

Response:

We have discussed the limitations of the 3D organoid model in disease modeling, including the specific point raised by the reviewer about the lack of recapitulating complex cell-cell interaction and brain structures (Page 18, Line 579).

Formatted: Font color: Text 1

6. Some missing references: “...Mao-B inhibitors or overexpression...for PD treatment” (p. 10); “neuronal

function was characterized...oligodendrocytes (29.6%)” (p. 17)

Response:

We thank reviewer for pointing out the error. We have inserted the citations in the revised manuscript.

Formatted: Font color: Text 1

Formatted: Font color: Text 1

Formatted: Font color: Text 1

7. Parkin R275W is incorrectly stated as PINK1 R275W (p. 9)

Response:

We corrected the error in the revised manuscript.

Formatted: Font color: Text 1

8. The manuscript is peppered with typographic errors. For example, “PRAK7” instead of “PARK 7” (p. 8); “glucocerebrocidas” instead of “glucocerebrocidase” (p. 10)

Response:

We thank reviewer for pointing out these typographic errors. We have corrected the errors accordingly and carefully checked for others in the revised manuscript.

Formatted: Font color: Text 1

Formatted: Font color: Text 1

Formatted: Font color: Text 1

Formatted: Font color: Text 1